# Validation and Psychometric Properties of the Spanish Version of the Hopkins Symptom Checklist-25 Scale for Depression Detection in Primary Care

**DOI:** 10.3390/ijerph18157843

**Published:** 2021-07-24

**Authors:** María Rodríguez-Barragán, María Isabel Fernández-San-Martín, Ana Clavería-Fontán, Susana Aldecoa-Landesa, Marc Casajuana-Closas, Joan Llobera, Bárbara Oliván-Blázquez, Eva Peguero-Rodríguez

**Affiliations:** 1Primary Health Centre La Mina, Gerència Territorial de Barcelona, Institut Català de la Salut, 08930 Barcelona, Spain; 2Faculty of Medicine, Autonomous University of Barcelona, 08193 Bellaterra, Spain; mcasajuana@idiapjgol.info; 3Institut Universitari d’Investigació en Atenció Primària Jordi Gol, 08007 Barcelona, Spain; mifsanmartin.bcn.ics@gencat.cat (M.I.F.-S.-M.); pegueroeva@gmail.com (E.P.-R.); 4Gerència Territorial de Barcelona, Institut Català de la Salut, 08007 Barcelona, Spain; 5I-Saúde Group, Galicia Sur Health Research Institute SERGAS-UVIGO, Servicio Galego de Saúde, 36201 Galicia, Spain; anaclaveriaf@gmail.com (A.C.-F.); susialdecoa@gmail.com (S.A.-L.); 6Primary Care Prevention and Health Promotion Research Network RedIAPP, 28001 Madrid, Spain; jllobera@ibsalut.caib.es (J.L.); bolivan@unizar.es (B.O.-B.); 7Primary Health Centre Beiramar, Área de Xestión Integrada de Vigo, SERGAS, 36201 Vigo, Spain; 8Institut d’Investigació Sanitària Illes Balears, 07120 Palma, Spain; 9Primary Care Research Unit of Mallorca, Balearic Islands Health Service, 07120 Palma, Spain; 10Department of Psychology and Sociology, University of Zaragoza, 50009 Zaragoza, Spain; 11Primary Health Centre El Castell, Institut Català de la Salut, 08860 Barcelona, Spain; 12Faculty of Medicine, University of Barcelona, 08007 Barcelona, Spain

**Keywords:** depression, depressive disorder, primary healthcare, family practice, general practitioners, anxiety, questionnaires, psychometrics

## Abstract

Depression constitutes a major public health problem due to its high prevalence and difficulty in diagnosis. The Hopkins Symptom Checklist-25 (HSCL-25) scale has been identified as valid, reproducible, effective, and easy to use in primary care (PC). The purpose of the study was to assess the psychometric properties of the HSCL-25 and validate its Spanish version. A multicenter cross-sectional study was carried out at six PC centers in Spain. Validity and reliability were assessed against the structured Composite International Diagnostic Interview (CIDI). Out of the 790 patients, 769 completed the HSCL-25; 738 answered all the items. Global Cronbach’s alpha was 0.92 (0.88 as calculated for the depression dimension and 0.83 for the anxiety one). Confirmatory factor analysis (CFA) showed one global factor and two correlated factors with a correlation of 0.84. Area under the curve (AUC) was 0.89 (CI 95%, 0.86–0.93%). For a 1.75 cutoff point, sensibility was 88.1% (CI 95%, 77.1–95.1%) and specificity was 76.7% (CI 95%, 73.3–79.8%). The Spanish version of the HSCL-25 has a high response percentage, validity, and reliability and is well-accepted by PC patients.

## 1. Introduction

Depression is a common condition among adults and can lead to harmful consequences. Worldwide, it is considered the third cause of years lost to disability [1], with a prevalence that increased by 17.8% between 2005 and 2015 [2]. In Europe, studies carried out in primary care (PC) settings have reported an incidence from 9.6% [3] to 20.2% [4]. The prevalence of depression in Spain is higher than the European mean and is associated with a negative perception of physical health, the presence of two or more difficulties in daily living activities, female gender [5], and some physical comorbidities [6].

Several instruments have been designed to screen mental disorders. As a collaborative international project, the Family Practice Depression and Multimorbidity (FPDM) group from the European General Practice Research Network (EGPRN) aimed to select a questionnaire to detect depression symptoms in PC patients [7]. Firstly, a systematic review of validated scales for screening and diagnosis of depression in adults was performed. Scales that had been compared to a psychiatric interview based on the Diagnostic and Statistical Manual of Mental Disorders (DSM) criteria with quantitative results and with participation of PC professionals were analyzed [8]. As a result of this systematic review, seven scales were identified: Geriatric Depression Scale of five items (GDS-5), Geriatric Depression Scale of 15 items (GDS-15), Geriatric Depression Scale of 30 items (GDS-30), Hospital Anxiety Depression Scale (HADS), Center for Epidemiologic Studies Depression Scale-Revised (CESD-R), Physical Symptom Checklist of 51 items (PSC-51), and Hopkins Symptom Checklist of 25 items (HSCL-25).

Secondly, the HSCL-25 was selected by consensus [9]. Validity, efficacy, and reproducibility were analyzed as quantitative criteria. Characteristics such as being a self-administered questionnaire, easiness of completion for patients, and the simplicity of its interpretation were taken into account to assess the ergonomics. The HSCL-25 is suitable for use in PC because of its high validity and reliability; moreover, its ergonomics make it easy to use for patients [9]. It is a self-report questionnaire designed to measure psychological distress based on the SCL-90 [10], a longer checklist designed by Derogatis et al. The full version of the SCL-90 covers nine symptom dimensions, with 25 items belonging to the anxiety and depression ones.

Thirdly, the questionnaire was translated into 13 European languages [11], including the Spanish version [12]. The translation and adaptation process consisted of an initial forward translation, a pilot study based on the Delphi methodology with the participation of family doctors, and a back translation. Comprehension analysis was carried out through cognitive debriefing in a sample of PC patients. At the last step, transcultural harmonization was performed simultaneously with other versions of the scale in different European languages [11].

Finally, validation of the different versions is in process, the French version has already been validated [13] and the Croatian one is under way.

Instruments should be tested and validated in different languages, cultures, settings, and populations in order to make comparisons and establish efficacy. The Consensus-based Standards for the selection of health Measurement Instruments (COSMIN) [14] initiative has developed criteria to evaluate the measurement properties of outcome measurement instruments. In addition, a considerable number of studies have assessed the HSCL-25 psychometric properties in various populations [15,16,17,18,19], including PC patients [20,21]. PC is the gateway to the healthcare system for most of the Spanish population. It is the ideal setting to study the prevalence of the most common diseases.

The purpose of this study was to assess the HSCL-25 psychometric properties and validate the scale’s Spanish version in a PC population.

## 2. Methods

### 2.1. Study Design

A cross-sectional multicenter design was used. The participants were patients attending primary healthcare centers (PHC) in Aragon (1), the Balearic Islands (1), and Galicia (4) taking part in the EIRA study [22]. Ethical approval was given by the Ethics Committee of Institut Universitari d’Investigació en Atenció Primària (IDIAP) Jordi Gol (reference number P16/025).

### 2.2. Participants

The selection criteria were those employed in the EIRA study. Eligible participants were patients aged between 45 and 75 years who had two or more of the following unhealthy behaviors: tobacco use, low adherence to the Mediterranean dietary pattern, and insufficient physical activity. Exclusion criteria were advanced serious illness, cognitive impairment, dependence in basic everyday activities, severe mental illness, inclusion in a long-term home healthcare program, treatment for cancer, end-of-life care, or no plan to reside in the area during the intervention period.

### 2.3. Recruitment and Sample Size

Recruitment was made by consecutive sampling of patients meeting the selection criteria and attending the PHC for any reason. The recruitment period took 6 months during 2017.

The COSMIN guide [23] was followed to calculate the sample size. It states that seven completed questionnaires are needed per each item of the scale and that at least 100 completed questionnaires are required to assess psychometric properties. As the HCSL-25 has 25 items, and taking into account a 10% possibility of missing values, 193 patients were needed to complete the questionnaire.

In order to estimate the sample size required to compare the HSCL-25 with the Composite International Diagnostic Interview (CIDI), the receiver operating curve (ROC) and the corresponding area under the curve (AUC) were calculated with the BIOSOFT application (http://www.biosoft.hacettepe.edu.tr/easyROC/, accessed on 31 January 2021) employing the following parameters:A 0.742 AUC to identify cases of depression (according to Nettelbladt et al. [20]).A type I error of 5% and a power of 95% were selected. Thus, 87 cases and 174 controls were needed (new AUC test: 0.80, standard AUC test: 0.74; case/control ratio: 2).

Taking into account that the estimated prevalence in PC is 16.3% [4], 533 patients were required to complete the scale to obtain 87 cases. 

To evaluate test–retest reliability, the same considerations and a 20% possibility of missing values were taken into account, 26 patients were needed to reach an acceptable correlation coefficient of 0.7 [24]. All the included patients were invited to participate in the telephonic retest.

### 2.4. Variables

Sociodemographic data (sex, age, nationality, marital status, current employment, and education level) were gathered from the participants. They were asked to complete the self-administered HSCL-25 questionnaire and other forms related to the EIRA study. Afterwards, trained professionals blinded to the HSCL-25 results conducted the CIDI with all the participants. Training consisted of a global presentation of the procedure of the interview, the reading question by question, role-playing with the interviewers, and resolution of doubtful situations. Retest of the HSCL-25 was telephonic to facilitate the viability; it was carried out between 1 and 3 months later.

### 2.5. Hopkins Symptom Checklist-25 (HSCL-25)

The HSCL-25 is a self-administered questionnaire that takes from five to ten minutes to complete [13]. It consists of 25 items on a four-point Likert scale: 1 = “Not at all,” 2 = “A little,” 3 = “Quite a bit,” 4 = “Extremely”. The tool has two well-known dimensions: items 1 to 10 belong to the anxiety dimension, whereas items 11 to 25 constitute the depression one. The HSCL-25 score is calculated by dividing the total score of items by the number of items answered, so the final score can range from 1 to 4. A cutoff value of 1.75 is generally used for diagnosis of major depression, defined as “a case in need of treatment”. This cutoff point is recommended as a valid predictor of mental disorder [15,17,25]. Our study was carried out using the Spanish version of the HSCL-25 obtained by means of translation and cultural adaption of the original English version [12].

### 2.6. Composite International Diagnostic Interview (CIDI)

The CIDI is a standardized structured diagnostic interview created by the World Health Organization (WHO) according to the DSM-IV and International Classification of Diseases (ICD-10) definitions and criteria. Used by trained interviewers for mental disorder assessment in the general population [26], it has demonstrated high validity and reliability [27]. Whilst the original CIDI was in English, it has been adapted into and validated for many languages using a common procedure overseen by the WHO [28]. Questions related to depression symptoms can be found in section E of the CIDI. In this study, it was considered the gold standard to assess the HSCL-25.

### 2.7. Patient Health Questionnaire (PHQ)

The PHQ is a well-known self-administered questionnaire used for common mental disorders. The PHQ-9 is the depression module in which each of the nine items is rated with a Likert scale that ranges from 0 to 3 [29]. The total score can vary from 0 to 27. Scores of 15 or more indicate major depression. For this study, the validated Spanish version [30] was employed.

### 2.8. Statistical Analysis

Analysis was conducted using STATA version 15 (manufacturer StataCorp LLC, Texas, USA).

#### 2.8.1. Missing Data

The missing values for scale item responses were imputed with the mean of the responses to the rest of the scale items of each individual (the participant’s most representative value). The subjects with less than 50% response were excluded. The same imputation was carried out for the retest values.

#### 2.8.2. Responding Process and Item Analysis

An analysis of the responding process was performed, looking for patterns of non-response and frequency response distribution of the items by category and sex. The discriminatory capacity of the items was assessed by comparing the two extreme groups. The discrimination index (DI) of each item was calculated by the mean difference of each group. Given that the response options have four possible categories, the DI could vary from −3 to +3.

#### 2.8.3. Internal Structure

Confirmatory factor analysis (CFA) was carried out based on the structure of the original English version. The factorial loads for the two models with only one factor and for the two correlated ones (anxiety and depression) were calculated. The robust maximum likelihood mean adjusted method was employed to carry out factorial analysis of the standardized values. To evaluate the estimated model fit, the absolute fit index was calculated with chi-squared distribution. Given that this value may be affected by the sample size, complementary indices were employed, including the root mean square error of approximation (RMSEA), the standardized root mean square residual (SRMR), and the coefficient of determination (CD). In addition, comparative indices such as the comparative fit index (CFI) and the Tucker–Lewis fit index (TLI) were employed.

#### 2.8.4. Criterion Validity

Criterion validity was measured by calculating the ROC curve for the HSCL-25 scale in comparison with the gold-standard CIDI. The AUC was estimated with 95% CI. Sensitivity, specificity, positive and negative predictive values, Youden index, and the best cutoff point were also assessed.

Concordance with the PHQ-9 was measured with the Pearson correlation coefficient and the prevalence- and bias-adjusted kappa, taking into account cutoff points of 1.75 and 15 for the HSCL-25 and PHQ-9, respectively.

#### 2.8.5. Internal Consistency

The contribution of the items to the internal consistency was analyzed with indicators based on correlation (homogeneity), covariance (Cronbach’s alpha coefficient), and regression (R^2^). The total Cronbach’s alpha and one for each of the two subscales were calculated. The value ≥0.7 was considered adequate [24].

#### 2.8.6. Test–Retest Reliability

Test–retest reliability was assessed by calculating the intraclass correlation coefficient (ICC) by the use of the mean of two evaluations (test and retest), absolute agreement, and a two-way mixed-effects model. 

## 3. Results

### 3.1. Participants

A total of 790 patients were selected for the HSCL-25 and 768 patients completed it (97.2% response rate). The participants’ mean age was 58.4 years (± 8.2) without significant gender differences; 54.4% were women. Table 1 depicts the sociodemographic characteristics of the sample. Women and men differed in marital status and current employment.

### 3.2. Responding Process and Item Analysis

Of the 23 participants excluded from the analysis (2.9%), 22 did not answer any of the items and one only responded to 12 of the 25 items. Thirty participants (3.8%) left between one and five items blank; these missing values were imputed. No non-response patterns were found; Appendix A shows the non-response patterns in detail.

The mean score of the items, the response percentages for each category, and the DI are depicted in Table 2. Item 20 “Worrying too much” with a mean of 2.14 had the highest global rating; it was followed by item 4 “Nervousness” (mean = 2.03). In contrast, item 18 “Thinking of ending one’s life” (mean = 1.09) had the lowest value, followed by item 9 “Feeling panic” (mean = 1.17). Women scored higher in all the items. The greatest difference between genders was observed in item 14 “Losing sexual interest” with a statistically significant difference of 0.73. The item that varied the least between genders was item 24 “Poor appetite,” and it was non-significantly different. 

With respect to response frequency distribution, 60.9% of the responses were found in the lowest response category “Not at all,” with the rating of 1. A floor effect was observed in item 18 “Thinking of ending one’s life,” with 93.6% of responses in the lowest category. None of the items presented a ceiling effect. 

The discrimination capacities of the items all showed a positive DI. Item 4 “Nervousness” discriminated the best with the DI of 1.43. In contrast, the item with the worst discrimination values was item 18 “Thinking of taking one’s life” (DI = 0.22).

### 3.3. Internal Structure: Confirmatory Factorial Analysis

The Satorra–Bentler comparative fit index was significant. Globally, the other indices showed that the proposed one-factor and two-correlated-factor models were reasonably acceptable. Table 3 depicts the fit indices for each model.

Table 4 shows the factor loading for each model and correlation in the two-factor model. All the factor loadings were positive and statistically significant (*p* < 0.001) and ≥ 0.30. Only item 24 “Poor appetite” had a loading below 0.4. With respect to the two correlated factors, the standardized values ranged from 0.3 for item 24 and 0.84 for item 17 “Feeling blue,” both in the depression category.

### 3.4. Criterion Validity: Relationship with the Gold-Standard CIDI

Of the 767 patients who completed the HSCL-25 scale, 736 also participated in the CIDI interview (96.0%). The 31 patients who did not take part in the interview were excluded from the following analysis. According to the CIDI, the global depression prevalence was 8.0% (CI 95%, 6.2–10.2%): 4.7% (CI 95%, 2.7–7.5%) in men and 10.8% (CI 95%, 8.0–14.4%) in women. With respect to the HSCL-25, the global prevalence was 28.5% (CI 95%, 25.3–31.9%) for the 1.75 cutoff point. Table 5 shows the different indices and values globally and by sex. The differing optimum cutoff points for women (1.76) and men (1.84) are noteworthy. Sensitivity was similar for both genders whilst specificity was better in men. The global ROC curves are depicted in Figure 1, by gender—in Figure 2. The global AUC was 0.892 (CI 95%, 0.856–0.928); in the gender analysis, it was greater in men. The optimum cutoff point for the Spanish version of the HSCL-25 was 1.76, with the Youden index of 64.8%.

### 3.5. Criterion Validity: Relationship with PHQ-9 External Criteria

The HSCL-25 scale and the PHQ-9 were completed by 761 patients. The Pearson coefficient for the values of both scales was 0.780 (CI 95%, 0.750–0.806). Considering both variables as categorical with cutoff points of 1.75 and 15 for the HSCL-25 and the PHQ-9, respectively, the prevalence- and bias-adjusted kappa (PABAK) value was 0.553, with the global agreement of 77.7% (CI 95%, 74.6–80.5%).

### 3.6. Reliability: Internal Consistency and Test–Retest Reliability

Intercorrelation between the items can be observed in Appendix A. All the correlations were positive and of a moderately low magnitude. The correlation mean was 0.362, with the standard deviation (SD) of 0.210. The intercorrelation range was 0.090–0.621. The correlation means and SD for the anxiety and depression subscales were 0.450 (SD, 0.278) and 0.419 (SD, 0.243), respectively.

Cronbach’s alpha coefficient for the total values and for each subscale, the total item correlations and determination coefficients (R^2^) are depicted in Table 6. The value obtained for the coefficient without the item was also calculated as shown in the middle column in Table 6. All the results were lower with the exception of item 24 “Poor appetite,” the elimination of which resulted in an increase in the global coefficient from the 4th decimal. Assessing the item-total correlation and the R^2^ of this item resulted in lower values in both cases as the item was the least consistent one.

The most homogeneous item was 17 “Feeling blue;” when eliminated, the internal consistency of the scale decreased to the lowest value; this item presented the highest item-total correlation and R^2^.

The telephone retest was completed by 94 participants. Test–retest reliability was 0.92 (CI 95%, 0.87–0.95), calculated with CCI. 

## 4. Discussion 

A major finding of our study is that the Spanish version of the HSCL-25 is an instrument with good acceptability and high response rate for PC patients. Its reliability in measuring depression is robust and presents considerable sensitivity and specificity when compared to the CIDI interview. The CFA demonstrated that the Spanish version is similar to the original English one.

For most of the Spanish population, PC consultations are the gateway to the healthcare system. Due to the high prevalence of depression [5], it is crucial that easy to use viable tools are available for the PC environment. As the HSCL-25 meets such characteristics [9], awareness of its psychometric properties is relevant, in particular, of those items that most contribute to detecting symptoms and thus permit discrimination between the healthy populations and the potentially depressed ones. In addition, PC professionals should be informed of the reliability of the scale and its sensitivity and specificity values which are key in order to establish its diagnostic utility. 

The study participants were PC patients aged 45–75 years who had taken part in the more extensive EIRA study [22]. Whilst this implied a restricted age range, which might signify a limitation, the sample was considered sufficiently representative of such individuals. The sample size was greater than the minimum required for the analysis according to the COSMIN guidelines [23], which are taken as reference in the field of psychometry. The statistical analysis was carried out based on the same recommendations. The content validity of the Spanish version of the HSCL-25 had been previously evaluated when it had been translated and transculturally adapted to Spanish and other official languages of the country [12].

With respect to item analysis, a considerable percentage of responses was available, and no definite pattern was observed. As a consequence, the questionnaire appears to be widely accepted by PC patients without any items which may cause discomfort or difficulty in understanding. As the study was carried out with patients attending the PHC for any reason, a high percentage of low-rating responses for the categories was expected. In addition, a floor effect was foreseen for item 18 “Thinking of ending one’s life” which concerned suicidal ideation. Taking into account the definition of depression according to the DSM-5 [31], it is not surprising that the item that best discriminated between the healthy population and the one with depressive symptoms referred to sadness. Item 17 “Feeling blue” was shown to be the most homogenous in all the analyses, with the highest correlation compared to the other scale items. It presented the highest coefficient of determination (that is to say, it could be predicted from the rest of the items) and most contributed to augmenting internal global consistency.

Regarding analysis of the scale’s factorial structure, this was performed with the CFA as the HSCL-25 has been widely studied with one single factor or two correlated ones even though other models have been proposed [15,32,33]. The fit indices for both models were acceptable, and the results indicated moderately elevated factorial loads. In the study of the two-factor model, there was a factorial correlation of 0.84 which indicated that the depression and anxiety dimensions strongly correlated in a positive manner. Such a figure is higher than that detected in previous studies [15]. The correlation is understandable as symptoms of anxiety are often observed in patients diagnosed with depression; moreover, anxiety and depression are frequently found to be associated comorbidities [34,35].

In other studies which compared the HSCL-25 scale with structured psychiatric interviews, a subsample of participants was selected for the latter to facilitate viability [13,16,17,20]. A strength of our work is that all the participants who responded to the HSCL-25 scale also took part in the structured CIDI interview imparted by trained professionals. We obtained 736 patients who fully answered both the scale and the gold-standard CIDI. Validity criteria were considerable, the global AUC was 0.890 (higher in men than in women). The global sensitivity and specificity by gender were elevated. The former was greater than that found in previous studies [13,16,17,20,25], whilst the latter was similar to the 73% reported by Nettelbladt et al. [20] and the 78% observed by Lundin et al. [16], both in Swedish populations. Other authors have described higher values [13,17]. In spite of the augmented number of false positives obtained, in a similar manner to other studies [36], the negative predictive value was greater than 97% for both genders. Such a finding indicates that the scale is a good tool for depression screening. With respect to the optimum cutoff point, both the global figure and the one for women were very similar to the 1.75 proposed in the original version and employed in other studies [21,37]. Nevertheless, 1.84 for men was higher, and contrasted with the findings of other authors where the cutoff point was greater for women [25]. When contrasting the total rating of the HSCL-25 scale with that of the PHQ-9 [29,38] of depression, an elevated correlation was obtained, and the PABAK was acceptable. Such an analysis reinforces the elevated criteria validity found. 

For the one-factor HSCL-25 scale, Cronbach’s alpha coefficient was 0.92, similar to the 0.93 obtained in the French version [13]. Nunally et al. established the critical level of reliability at 0.70; they stated, however, that for the key individual decisions, such as the diagnosis of depression, reliability should be raised to 0.90 [39]. Cronbach’s alpha for the subscales of depression and anxiety taken separately was greater than 0.80. Such findings demonstrate the strong reliability of the scale to measure depression, especially when employed as a single dimension instrument. The test–retest reliability was greater than 0.90, higher than that observed in other studies [18], which indicates that the ratings are stable over time. The time interval between the baseline interview and the retest was considered adequate, the test conditions—acceptable, in spite of the retest being carried out by telephone to avoid overwhelming the participants.

Other shortened versions of the scale with five and 10 items [40,41] presenting acceptable reliability have been proposed. They could be of use, taking into account the characteristics of the PC environment. These studies have been performed in other languages and it might be of interest to translate them into Spanish.

Our findings indicate that, in the future, the Spanish version of the HSCL-25 scale could be employed as a diagnostic tool for depression in PC consultations. Our study has taken place within the framework of a European project [8,9], in which a common methodology has been used for the translation and adaptation of different languages. We believe, therefore, that the HSCL-25 is a good tool to carry out research concerning the prevalence of depression at the European level once the various language versions have been validated.

## 5. Conclusions

The Spanish version of the Hopkins Symptom Checklist-25 is well-accepted by patients and shows high validity and reliability to detect depression symptoms in primary care. It has a similar factorial structure to the original English version and can be used in daily practice and for research.

## Figures and Tables

**Figure 1 ijerph-18-07843-f001:**
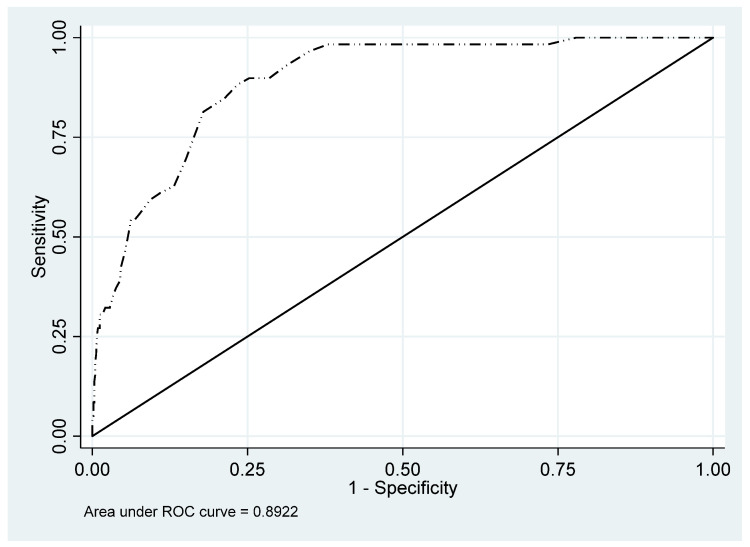
Total ROC (receiver operating curve) and AUC.

**Figure 2 ijerph-18-07843-f002:**
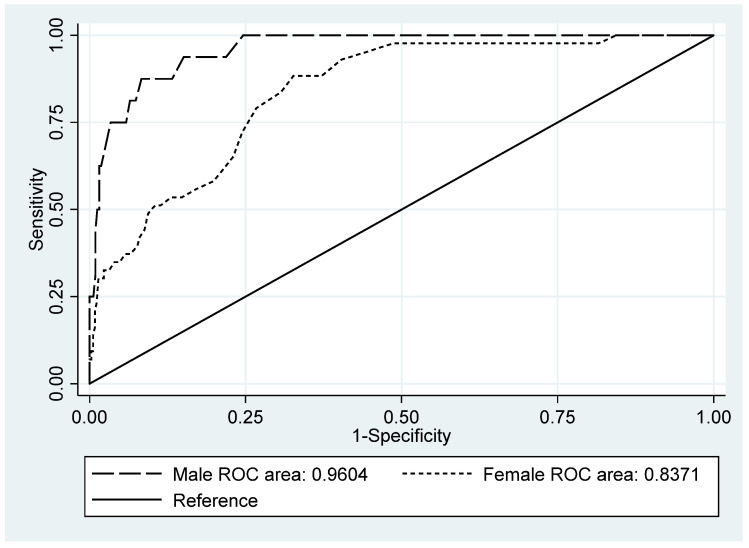
ROC curves and AUC by gender.

**Table 1 ijerph-18-07843-t001:** Sociodemographic characteristics of the sample.

Sociodemographic Variables	Male		Female		Total		*p*
	n	%	n	%	n	%	
	350	45.6	417	54.4	767		
**Marital status**							
Married/with a partner	259	74.0	291	69.8	550	71.7	0.001
Single	31	8.9	25	6.0	56	7.3	
Separated/divorced	48	13.7	55	13.2	103	13.4	
Widow(er)	11	3.1	46	11.3	57	7.4	
Others (religious…)	1	0.3	0	0	1	0.1	
**Education**							
Secondary or higher	178	50.9	190	45.6	368	48.0	0.153
Primary or lower	172	49.1	226	54.2	398	51.9	
No response	0	0	1	0.2	1	0.1	
**Current employment**							
Employed	153	43.7	160	38.4	313	40.8	<0.001
Housewife	2	0.6	109	26.1	111	14.5	
Unemployed	45	12.9	38	9.1	83	10.8	
Retired	128	36.6	81	19.4	209	27.3	
Others (student, sick leave, disability)	20	5.7	28	6.7	48	6.3	
No response	2	0.6	1	0.2	3	0.4	

**Table 2 ijerph-18-07843-t002:** Item analysis: missing values, response distribution, mean score, and discrimination index.

Item	Response Values (*n* = 790)					Score							DI
	Missing Values	Not at All	A Little	Quite a Bit	Extremely	Male (*n* = 350)		Female (*n* = 417)		Total (*n* = 767)		*p*	
	%	%	%	%	%	Mean	SD	Mean	SD	Mean	SD		
1. Being scared for no reason	2.8	75.6	18.1	4.8	1.3	1.14	0.41	1.46	0.73	1.32	0.63	<0.001	0.61
2. Feeling fearful	3.3	67.5	25.0	5.7	1.7	1.27	0.51	1.54	0.77	1.42	0.68	<0.001	0.77
3. Faintness	3.2	54.4	35.1	8.5	2.1	1.47	0.65	1.68	0.79	1.58	0.73	<0.001	1.01
4. Nervousness	2.9	31.8	40.0	21.1	7.0	1.81	0.84	2.23	0.91	2.03	0.90	<0.001	1.43
5. Heart racing	3.2	62.1	29.3	6.3	2.4	1.36	0.60	1.59	0.79	1.49	0.72	<0.001	0.84
6. Trembling	3.2	76.5	18.0	3.9	1.6	1.28	0.58	1.33	0.65	1.31	0.62	0.253	0.59
7. Feeling tense	3.4	42.5	39.6	12.7	5.2	1.64	0.75	1.94	0.91	1.81	0.85	<0.001	1.26
8. Headache	3.3	59.3	26.7	9.9	4.0	1.43	0.72	1.72	0.89	1.59	0.83	<0.001	0.77
9. Feeling panic	3.2	87.0	9.7	2.5	0.9	1.13	0.43	1.21	0.55	1.17	0.50	0.046	0.39
10. Feeling restless	2.9	40.8	44.5	11.9	2.9	1.67	0.72	1.85	0.80	1.77	0.77	<0.001	1.10
11. Feeling low in energy	3.0	37.6	40.8	15.1	6.5	1.70	0.75	2.08	0.95	1.91	0.88	<0.001	1.34
12. Blaming oneself	3.0	61.2	27.5	8.0	3.4	1.43	0.65	1.62	0.87	1.54	0.78	<0.001	0.94
13. Crying easily	3.3	48.2	31.0	12.8	8.0	1.57	0.80	2.00	1.01	1.80	0.94	<0.001	1.04
14. Losing sexual interest	3.3	48.9	24.0	14.2	12.9	1.51	0.82	2.24	1.14	1.91	1.07	<0.001	1.29
15. Feeling lonely	2.9	65.6	23.3	6.1	5.0	1.32	0.66	1.66	0.90	1.50	0.82	<0.001	1.02
16. Feeling hopeless	3.3	75.9	16.0	4.7	3.4	1.23	0.57	1.46	0.82	1.36	0.73	<0.001	0.82
17. Feeling blue	3.2	52.3	34.4	8.6	4.7	1.40	0.66	1.87	0.89	1.66	0.82	<0.001	1.31
18. Thinking of ending one’s life	3.2	93.6	4.4	1.7	0.3	1.06	0.31	1.11	0.40	1.09	0.36	0.041	0.22
19. Feeling trapped	3.0	72.9	17.3	6.8	3.0	1.32	0.65	1.47	0.81	1.40	0.75	<0.001	0.88
20. Worrying too much	3.2	27.9	39.0	24.6	8.5	1.95	0.83	2.29	0.96	2.14	0.92	<0.001	1.29
21. Feeling no interest	3.5	79.1	14.6	4.3	2.0	1.20	0.51	1.37	0.72	1.29	0.64	<0.001	0.69
22. Feeling that everything is an effort	2.9	48.6	36.9	9.8	4.7	1.56	0.67	1.83	0.92	1.71	0.83	<0.001	1.18
23. Worthless feeling	2.8	81.4	12.1	4.7	1.8	1.17	0.51	1.35	0.71	1.27	0.63	<0.001	0.59
24. Poor appetite	2.9	83.3	11.2	3.7	1.8	1.23	0.58	1.25	0.63	1.24	0.60	0.723	0.43
25. Sleep disturbance	2.9	47.5	28.2	13.6	10.8	1.74	0.93	2.00	1.07	1.88	1.01	<0.001	1.21
**Total**	3.1	60.9	25.9	9.0	4.2	1.42	0.64	1.69	0.82	1.57	0.76	<0.001	

DI: discrimination index.

**Table 3 ijerph-18-07843-t003:** Fit indices in the studied factor models.

Model	X^2^_SB (df)	*p*	CFI_SB	TLI_SB	RMSEA (90% CI)	*p* Close	RMSEA_SB	SRMR	CD
One factor	1600.3 (275)	<0.001	0.828	0.812	0.079 (0.076–0.083)	<0.001	0.061	0.059	0.931
Two correlated factors	899.5 (274)	<0.001	0.862	0.849	0.072 (0.068–0.076)	<0.001	0.055	0.055	0.970

X^2^_SB: Satorra–Bentler chi-squared statistic; df: degrees of freedom; CFI_SB: Satorra–Bentler comparative fit index; TLI_SB: Satorra–Bentler Tucker–Lewis fit index; RMSEA: root mean square error of approximation; RMSEA_SB: Satorra–Bentler root mean square error of approximation; SRMR: standardized root mean square residual, CD: coefficient of determination.

**Table 4 ijerph-18-07843-t004:** Confirmatory factorial analysis: factor loading values and correlation between two depression and anxiety factors.

Item	One Factor	Two Correlated Factors	
		Anxiety	Depression
1. Being scared for no reason	0.45	0.49	
2. Feeling fearful	0.52	0.56	
3. Faintness	0.60	0.59	
4. Nervousness	0.67	0.75	
5. Heart racing	0.53	0.57	
6. Trembling	0.43	0.48	
7. Feeling tense	0.68	0.74	
8. Headache	0.39	0.41	
9. Feeling panic	0.40	0.44	
10. Feeling restless	0.65	0.70	
11. Feeling low in energy	0.67		0.66
12. Blaming oneself	0.57		0.58
13. Crying easily	0.51		0.50
14. Losing sexual interest	0.51		0.52
15. Feeling lonely	0.65		0.68
16. Feeling hopeless	0.61		0.65
17. Feeling blue	0.82		0.84
18. Thinking of ending one’s life	0.43		0.44
19. Feeling trapped	0.60		0.62
20. Worrying too much	0.58		0.56
21. Feeling no interest	0.62		0.65
22. Feeling that everything is an effort	0.68		0.69
23. Worthless feeling	0.53		0.56
24. Poor appetite	0.30		0.30
25. Sleep disturbance	0.48		0.47
**Factor correlation**		0.84	

**Table 5 ijerph-18-07843-t005:** Optimum cutoff points, global and gender-specific sensitivity, specificity, positive and negative predictive values in the HSCL-25 scale.

Calculated for	Index	Male (*n* = 341)			Female (*n* = 395)			Total (*n* = 736)		
		Value	CI 95%, Lower Limit	CI 95%, Upper Limit	Value	CI 95%, Lower Limit	CI 95%, Upper Limit	Value	CI 95%, Lower Limit	CI 95%, Upper Limit
Original cutoff, 1.75	Sensitivity	87.5	61.7	98.4	88.4	74.9	96.1	88.1	77.1	95.1
	Specificity	86.8	82.6	90.3	67.3	62.2	72.2	76.7	73.3	79.8
	PPV	24.6	14.1	37.8	18.3	12.5	25.4	24.8	19.1	31.2
	NPV	99.3	97.5	99.9	97.9	95.2	99.3	98.7	97.3	99.5
	AUC	0.960	0.927	0.994	0.837	0.781	0.893	0.892	0.856	0.928
	Optimal cutoff point	1.84			1.76			1.76		
Optimal cutoff point	Sensitivity	87.5	61.7	98.4	83.7	69.3	93.2	84.7	73.0	92.8
	Specificity	91.7	88.1	94.5	69.3	64.2	74.1	78.7	75.5	81.8
	PPV	34.1	20.1	50.6	25.0	18.2	32.9	25.8	19.8	32.5
	NPV	99.3	97.6	99.9	97.2	94.3	98.9	98.3	96.9	99.2

AUC: area under the curve. PPV: positive predictive value. NPV: negative predictive value.

**Table 6 ijerph-18-07843-t006:** Cronbach’s alpha coefficient of each item, globally and by subscales.

Item	Item-Total Correlation	Cronbach’s Alpha	R^2^
1. Being scared for no reason	0.4466	0.9148	0.3697
2. Feeling fearful	0.5140	0.9137	0.4154
3. Faintness	0.5724	0.9127	0.4640
4. Nervousness	0.6545	0.9109	0.5250
5. Heart racing	0.5172	0.9136	0.3657
6. Trembling	0.4226	0.9151	0.3413
7. Feeling tense	0.6536	0.9109	0.5220
8. Headache	0.3760	0.9163	0.2462
9. Feeling panic	0.3907	0.9156	0.2939
10. Feeling restless	0.6206	0.9117	0.4805
11. Feeling low in energy	0.6472	0.9110	0.5439
12. Blaming oneself	0.5454	0.9131	0.3385
13. Crying easily	0.4881	0.9145	0.2909
14. Losing sexual interest	0.4832	0.9153	0.2866
15. Feeling lonely	0.5973	0.9121	0.4857
16. Feeling hopeless	0.5728	0.9127	0.4842
17. Feeling blue	0.7799	0.9085	0.6677
18. Thinking of ending one’s life	0.4033	0.9159	0.2747
19. Feeling trapped	0.5577	0.9129	0.4085
20. Worrying too much	0.5592	0.9129	0.3770
21. Feeling no interest	0.5752	0.9129	0.4460
22. Feeling that everything is an effort	0.6432	0.9112	0.5494
23. Worthless feeling	0.4995	0.9140	0.3552
24. Poor appetite	0.3065	0.9167	0.1568
25. Sleep disturbance	0.4747	0.9152	0.2776
**Total**		**0.9166**	
**Anxiety subscale** (items 1–10)		**0.8306**	
**Depression subscale** (items 11–25)		**0.8784**	

R^2^: determination coefficient.

## Data Availability

All the principal investigators had access to the complete database. The datasets generated and analyzed during the study are available from the corresponding author.

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
