# Peer review of "Validation and Psychometric Properties of the Spanish Version of the Hopkins Symptom Checklist-25 Scale for Depression Detection in Primary Care"

_ijerph, 2021, doi:10.3390/ijerph18157843_

Round 1

Reviewer 1 Report

This study examined the psychometric properties of the Spanish version of the HSCL-25 to use for screening the individuals with depression in primary care units. The study procedure and statistical methods were reasonable.

I have only two minor concerns.

  1. “The full version of the HSCL-25 covers nine symptom dimensions, with 25 items belonging to the anxiety and depression ones.” As I know, the HSCL-25 contains only 25 items. Is the full version also named “HSCL-25”?
  2. Who conducted the diagnostic interviewing based on the Composite International Diagnostic Interview (CIDI)? What kinds of training did these interviewers have received?

Author Response

Dear reviewer, thank you for your comments, I proceed to respond to your questions:

“The full version of the HSCL-25 covers nine symptom dimensions, with 25 items belonging to the anxiety and depression ones.” As I know, the HSCL-25 contains only 25 items. Is the full version also named “HSCL-25”?

The full version of HSCL-25 is named Symptom Checklist-90 (SCL-90) and has nine dimensions. The HSCL-25 contains two of those nine versions, the depression and the anxiety ones.

The sentence in line 54 has been changed for a better understanding.

Who conducted the diagnostic interviewing based on the Composite International Diagnostic Interview (CIDI)? What kinds of training did these interviewers have received?

The CIDI was conducted by trained psychologists. The training consisted of a global presentation of the interview and specific of the depression section, professionals were trained on the procedure of the interview and the reading question by question. Finally, role-playing with the interviewers and resolution of doubtful situations were carried out.

A new sentence has been added to the text of the manuscript to clarify that aspect.

Kind regards,

María Rodríguez-Barragán

Reviewer 2 Report

Thank you to the editor and the authors for the opportunity to review this manuscript. The authors examine the psychometric properties of the Spanish version of widely used HSCL-25 scale. The authors gathered a large group of participants suitable for such validation and obtained a very good psychometric values of the questionnaire.

The article was written in a scientific way and contemplates the rules of the journal. The study is adequate to the scope of the journal in the area of theoretical article. The manuscript presents an important theme for the Psychology and Human Sciences area and problematizes about your research problem with a current literature.
The objectives are clearly stated and are effectively developed and answered in the text. The bibliography used is current and adequate and the introduction contemplates the importance of recent empirical studies addressing the HSCL-25 construction and its language versions. The results are in line with the objectives of the study and were presented in a clear manner. The discussion meets the objectives of the study and deepens the understanding of the data presented. The text meets the formal requirements of scientific language. However, some minor adjustments will be necessary.

Line 111: the authors do not describe the translation and adaptation process of the questionnaire. They only refer to their own article published in Spanish. It would be good if some information appeared in this section.

Line 175: What was the time interval between the measurements? How were 94 people selected to participate in the retest?

Author Response

Dear reviewer, thank you for your comments, I proceed to respond to your questions:

Line 111: the authors do not describe the translation and adaptation process of the questionnaire. They only refer to their own article published in Spanish. It would be good if some information appeared in this section.

The translation and adaptation process was carried out following the guidelines of the International Society for Pharmacoeconomics and Outcomes Research (ISPOR). It consisted of an initial forward translation, a pilot study based on Delphi methodology with the participation of family doctors and a back-translation. Equivalence analysis was carried out. Comprehension analysis was carried out through cognitive debriefing in a sample of Primary Care patients. Finally, transcultural harmonization was done simultaneously with other versions of the scale in different European languages.

A short explanation about the process of translation and adaptation has been added to the Introduction section in line 56.

Line 175: What was the time interval between the measurements? How were 94 people selected to participate in the retest?

The time between test and retest was between 1 and 3 months, this aspect is explained in line 109-110 (variables section). All participants were invited to do the telephonic retest, but only 94 participants accepted. We added this information in line 103 (sample size section).

Kind regards,

María Rodríguez-Barragán

Reviewer 3 Report

Overall this study is sound and informative, and the methodology section is well written. Some suggestions may apply to the manuscript before its publication.

In lines 45-47: The authors did an initial systematic review to identify questionnaires. The authors need to provide more detailed explanations about the review process. Plus based on DSM criteria, the questionnaires were examined. The authors need to provide why they selected DSM criteria and how it worked out.

In line 47, the authors said validity efficacy and reproducibility were analyzed. Please provide more explanations about that.

In line 48, the authors said other characteristics were considered. Please provide more explanations about that.

In line 51, the authors said its ergonomics make it is easy to use. It’s unclear and clarify that.

Overall the authors need to provide more explanations about how they selected HSCL-25 by consensus in the introduction.

In line 52-53, the authors may change the sentence to “It is a self report questionnaire designed to measure psychological distress based on the SCL-90, a longer checklist developed by Derogatis et al.

In line 55, the authors mentioned “secondly” but I can’t find firstly or first. Please edit that.

In line 64, “differing populations” may be changed to “various populations.”

In line 66, the authors mentioned “assess HSCL-25 in a PC population”. I know the authors briefly discussed why PC population is worth to be studied at the beginning of the introduction. But the authors need to provide more detailed explanations about why they selected the PC population in the introduction.

In line 82, the sentence needs to be rewritten.

In lines 96-98, the sentence needs to be rewritten.

In the methodology section, the authors may add exploratory factor analysis to identify factors.

Author Response

Dear reviewer, thank you for your comments, I proceed to respond to your questions:

In lines 45-47: The authors did an initial systematic review to identify questionnaires. The authors need to provide more detailed explanations about the review process. Plus based on DSM criteria, the questionnaires were examined. The authors need to provide why they selected DSM criteria and how it worked out.

In line 47, the authors said validity efficacy and reproducibility were analyzed. Please provide more explanations about that.

In line 48, the authors said other characteristics were considered. Please provide more explanations about that.

In line 51, the authors said its ergonomics make it is easy to use. It’s unclear and clarify that.

Overall the authors need to provide more explanations about how they selected HSCL-25 by consensus in the introduction.

Firstly, a systematic review of validated scales for screening and diagnose of depression in adults was done. Scales that had been compared to a psychiatric interview based on DSM criteria, with quantitative results and with participation of PC professionals were analysed. As a result of this systematic review, 7 scales were identified: Geriatric Depression Scale of 5 items (GDS 5), Geriatric Depression Scale of 15 items (GDS 15), Geriatric Depression Scale of 30 items (GDS 30), Hospital Anxiety Depression Scale (HADS), Center for Epidemiologic Studies Depression Scale-Revised (CESD-R), Physical Symptom Checklist of 51 items (PSC-51), and Hopkins Symptoms Checklist of 25 items (HSCL-25).

Secondly, the HSCL-25 was selected by consensus. Validity, efficacy and reproducibility were analysed as quantitative criteria. Characteristics such as being a self-administered questionnaire, easiness to fulfil it for patients and the simplicity of its interpretation were taken into account to assess the ergonomics.

Paragraphs in the introduction have been rewritten to provide more information about the previous methodology to select the HSCL-25.

In line 52-53, the authors may change the sentence to “It is a self report questionnaire designed to measure psychological distress based on the SCL-90, a longer checklist developed by Derogatis et al”.

This change has been done.

In line 55, the authors mentioned “secondly” but I can’t find firstly or first. Please edit that.

It has been corrected adding “firstly” at the beginning of the paragraph.

In line 64, “differing populations” may be changed to “various populations.”

This change has been done.

In line 66, the authors mentioned “assess HSCL-25 in a PC population”. I know the authors briefly discussed why PC population is worth to be studied at the beginning of the introduction. But the authors need to provide more detailed explanations about why they selected the PC population in the introduction.

PC is the gateway to the healthcare system in Spain for most of the Spanish population. PC is the ideal environment to screen and diagnose prevalent diseases, such as depression (as we know that it has a high prevalence among the population).

A sentence about this aspect has been added at the end of the Introduction section, in line 64.

In line 82, the sentence needs to be rewritten.

The sentence has been rewritten.

In lines 96-98, the sentence needs to be rewritten.

The sentence has been rewritten.

In the methodology section, the authors may add exploratory factor analysis to identify factors.

As the HSCL-25 has a well-known structure in its original version of one only factor and two correlated factors, we decided to conduct only the confirmatory factor analysis (CFA) to confirm that the Spanish version of HSCL-25 had the same internal structure than the original English version. As it is shown in the manuscript, our results confirm that all factor loadings were positive and statistically significant and ≥ 0.30. We consulted different experts in psychometry that agreed and recommended us to conduct the CFA.

Kind regards,

María Rodríguez-Barragán

Reviewer 4 Report

The authors investigated the psychometric properties and validation of the Spanish version of the Hopkins Symptom Checklist-25 scale for depression for its use in Primary Care settings.

The paper is well-written, interesting for the readers and useful for the identification of depressive symptoms.

I consider that the manuscript should be published in the journal. However, several minor changes should be made before publishing it.

In the introduction section, the authors should expand some paragraphs explaining why the current scales or the most useful scales are underused or should be improved with the use of another scale.

The authors, at the introduction, highlighted that the Hopkins Symptom Checklist-25 is easy to use for patients. How many time should they have? Is long to administrate? Further details should be provided.

Which kinf of differences may clinicians expect when comparing results of the self-administered HSCL-25 and the telephonic retest of HSCL-25? This should be explained in the methods section.

In the results section, women and men seem to differ in marital status and current employment. How do the authors interpret these results? Are they suggesting gender differences in "depressive-related symptoms"?

The discussion section is really good and well-written. The conclusions section is brief. Should the authors expand the conclusions section?

Author Response

Dear reviewer, thank you for your comments, I proceed to respond to your questions:

In the introduction section, the authors should expand some paragraphs explaining why the current scales or the most useful scales are underused or should be improved with the use of another scale.

Some paragraphs of the Introduction section have been rewritten to give additional information about how was the HSCL-25 selected and why were other scales rejected.

The authors, at the introduction, highlighted that the Hopkins Symptom Checklist-25 is easy to use for patients. How many time should they have? Is long to administrate? Further details should be provided.

The HSCL-25 takes from five to ten minutes to complete, this is explained in line 112-113 (HSCL-25 section in Methods). More information has been added to the Introduction section about the ergonomics of the HSCL-25.

Which kind of differences may clinicians expect when comparing results of the self-administered HSCL-25 and the telephonic retest of HSCL-25? This should be explained in the methods section.

The retest was telephonic to facilitate the viability of the study, as participants were asked for a lot of information in the first interview. This is a limitation. Although we might have expected differences because of that reason, we found a high test-retest reliability, so we can assume that the fact of doing the telephonic retest was not a problem for reliability.

In the results section, women and men seem to differ in marital status and current employment. How do the authors interpret these results? Are they suggesting gender differences in "depressive-related symptoms"?

We have detected statistically significant differences in marital status and current employment; there are more widow and housewife among the women. We also detected different score for HSCL-25 between male and female, women scored higher in all items. As many studies have reported, female gender is associated with more depressive-related symptoms. We are going to conduct further analysis to explore this gender differences, this results will be published in a future original article.

The discussion section is really good and well-written. The conclusions section is brief. Should the authors expand the conclusions section?

Thank you for your comment about the discussion section. We consider that all important aspects are reflected in the discussion section, so that no longer explanations need to be done in the conclusions section.

Kind regards,

María Rodríguez-Barragán